# Uncovering Causal Variables in Transformers using Circuit Probing

## Abstract

Neural network models have achieved high performance on a wide variety of complex tasks, but the algorithms that they implement are notoriously difficult to interpret. In order to understand these algorithms, it is often necessary to hypothesize intermediate variables involved in the network's computation. For example, does a language model depend on particular syntactic properties when generating a sentence? However, existing analysis tools make it difficult to test hypotheses of this type. We propose a new analysis technique – *circuit probing* – that automatically uncovers low-level circuits that compute hypothesized intermediate variables. This enables causal analysis through targeted ablation at the level of model parameters. We apply this method to models trained on simple arithmetic tasks, demonstrating its effectiveness at (1) deciphering the algorithms that models have learned, (2) revealing modular structure within a model, and (3) tracking the development of circuits over training. We compare circuit probing to other methods across these three experiments, and find it on par or more effective than existing analysis methods. Finally, we demonstrate circuit probing on a real-world use case, uncovering circuits that are responsible for subject-verb agreement and reflexive anaphora in GPT2 small and medium.

## 1 Introduction

Transformer models are the workhorse of modern machine learning, driving breakthroughs in subfields as disparate as NLP (Devlin et al., 2018; Radford et al., 2019; Brown et al., 2020), computer vision (Dosovitskiy et al., 2020), and reinforcement learning (Chen et al., 2021). Despite their success, little is known about the algorithms that they learn to implement. This central problem has inspired a flurry of analysis and interpretability research trying to "open the black box" (Rogers et al., 2021; Elhage et al., 2021; Belinkov, 2022). Despite considerable effort, these models remain almost entirely opaque.

One challenge that is inherent to interpreting a model that succeeds at a complex task is that researchers often do not have a complete picture of the algorithm that they are attempting to uncover. However, they may be able to propose high-level causal variables that are constituents of such an algorithm. For example, one may intuit that computing the syntactic number of the subject noun of a sentence might be useful for language modeling (Chomsky, 1965; Linzen et al., 2016). This variable must be causally implicated in an algorithm that solves the language modeling task, as it constrains the rest of the sentence due to agreement rules (i.e. the syntactic number of the main verb must match the syntactic number of the subject). However, it leaves open infinite possibilities in which other variables influence the prediction of the next token. Though we focus on language modeling, this discussion applies more generally to any complex domain where neural networks are applied, from vision Dosovitskiy et al. (2020) to astronomy (Ćiprijanović et al., 2020) to protein folding (Jumper et al., 2021).

We propose *circuit probing* to enable the investigation of intermediate causal variables in Transformers (Vaswani et al., 2017). Circuit probing introduces a trainable binary mask over model weights, which is optimized to uncover a circuit that computes a high-level intermediate variable (if one exists). This technique enables researchers to (1) test whether high-level intermediate variables are represented by the model, (2) test whether they are **causally** implicated in model behavior (rather

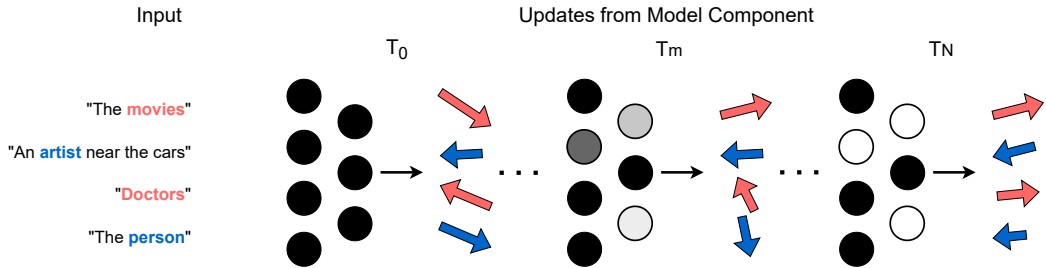

Figure 1: Schematic visualization of circuit probing for an intermediate variable representing the syntactic number of the subject of a sentence. Plural subjects are represented in red, singular subjects in blue. At step $T_0$, prior to training a binary mask, the model component (Attention block or MLP) produces residual stream updates (Elhage et al., 2021) that are not partitioned by syntactic number. Circuit probing optimizes a binary mask over model weights, resulting in a circuit whose updates are partitioned by syntactic number at step $T_N$.

than simply decodable from model representations), and (3) reveal the particular subset of model weights that compute it.

We first demonstrate this method on a series of simple arithmetic tasks, showing that it is *faithful* to the underlying model (i.e. it only provides evidence in support of causal variables that are actually represented by the model) and reveals circuits that are causally implicated in model behavior. We then use circuit probing to analyze two syntactic phenomena on GPT2 small and medium (Radford et al., 2019), uncovering particular model components responsible for subject-verb agreement and reflexive anaphora agreement[1].

**Limitations of Existing Causal Methods**   We compare circuit probing to two other techniques that seek to test for the existence of intermediate variables in neural networks: probing and causal abstraction analysis. Probing involves learning a classifier to decode information about a hypothesized intermediate variable from model activations (Tenney et al., 2019; Hewitt & Manning, 2019; Ettinger, 2020; Li et al., 2022; Nanda et al., 2023). Prior work has demonstrated that probing oftentimes mischaracterizes the underlying computations performed by the model (Hewitt & Liang, 2019; Zhang & Bowman, 2018; Voita & Titov, 2020)). Among other methods (Ravfogel et al., 2020; 2022; Belrose et al., 2023), counterfactual embeddings were introduced to fix this problem, enabling causal analysis using probes (Tucker et al., 2021). We empirically demonstrate that circuit probing is more faithful to a model's computation than linear or nonlinear probing (See Sections 3.2, 3.3), and that counterfactual embeddings do not always solve this problem (See Sections 3.1, 3.2).

Causal abstraction analysis intervenes on the activation vectors produced by Transformer layers to localize intermediate variables to particular vector subspaces (Geiger et al., 2021; 2023; Wu et al., 2023). However, causal abstraction analysis requires hypothesizing a complete causal graph – a high-level description of how inputs are mapped to predictions, including all interactions between intermediate variables. This is impossible for most real-world tasks on which we want to apply neural networks (such as language modeling). We empirically demonstrate the utility of recent causal abstraction analysis techniques when full causal graphs *are* available, and show that circuit probing arrives at the same results (See Sections 3.1, 3.2).

## 2   CIRCUIT PROBING

The key intuition behind circuit probing is that model components that compute an intermediate variable should produce outputs that are partitioned according to that variable. For example, if a model component is computing the syntactic number of the subject noun, then that component should produce outputs that fall into one of two equivalence classes, corresponding to singular subjects and plural subjects.

---

[1]We release our code at: https://anonymous.4open.science/r/Circuit_Probing-0672. Circuit probing is implemented using the NeuroSurgeon package (Lepori et al., 2023a).

The first step in circuit probing is to formalize the equivalence classes that we expect our intermediate variable to induce within a model component. To do so, we first label a dataset according to the intermediate variable that we are interested in. For example, singular-subject sentences are mapped to one label, and plural-subject sentences are mapped to another. Then, for a dataset of size $N$, we construct an $N$ by $N$ idealized similarity matrix, which maps pairs of inputs with the same label to 1 and pairs of inputs with different labels to 0. If the outputs produced by a model component share this pairwise similarity structure, then we can say that this model component computes (something isomorphic to) the intermediate variable that we are testing (Kriegeskorte et al., 2008). Circuit probing is illustrated in Figure 1.

It is unlikely that a model component will be fully dedicated to one hypothesized intermediate variable. Recent work has shown that neural networks often exhibit structure at the level of subnetworks (Csordás et al., 2020; Lepori et al., 2023b; Hod et al., 2021). Thus, we freeze the model's parameters and optimize a binary mask over neurons in the model (i.e., over columns in the matrices of the linear transformations that comprise MLP and attention blocks). Formally, given a model component $C$, with parameters $\theta$, we introduce a set of mask parameters $m$, where the values of $m$ are tied across weights in a neuron. Model weights are elementwise multiplied with mask parameters, which results in the circuit $C_{\theta \odot m}$. During training, we freeze $\theta$ and only optimize the parameters in $m$. We use continuous sparsification (Savarese et al., 2020), a model pruning technique that anneals a soft mask into a discrete mask over training, to learn $m$ (See Appendix B). Our optimization objective is soft nearest neighbors loss (See Appendix C), a contrastive loss that minimizes (cosine) distance between inputs of the same class, and maximizes (cosine) distance between inputs of different classes (Salakhutdinov & Hinton, 2007; Frosst et al., 2019). We also train with $l_0$ regularization to encourage sparse binary masks. If this process is successful, it results in a sparse circuit within a model component that computes the hypothesized intermediate variable.

We note that Transformers consist of a series of attention and MLP blocks that produce additive updates to the residual stream (Elhage et al., 2021). Circuit probing looks within these attention and MLP blocks to find circuits that compute high-level variables. In practice, we will search over all blocks iteratively.

## 3 EXPERIMENTS

We present four diverse experiments in order to illustrate the breadth of questions that circuit probing can help address. Experiments 1, 2, and 3 investigate toy models trained on simple arithmetic tasks, where full causal graphs are easy to construct. These experiments both (1) nuance or reproduce results from prior work and (2) contrast circuit probing with existing analysis methods. Experiment 4 applies circuit probing to language models to demonstrate that the method generalizes to a more realistic model and setting.

Experiment 1 uses circuit probing to distinguish between two algorithms that a Transformer might implement to solve an arithmetic task, and demonstrates that circuit probing agrees with all existing analysis methods in this simple setting. Experiment 2 investigates the internal structure of Transformer representations and demonstrates the superiority of circuit probing over nonlinear probing and causal abstraction analysis for this purpose. Experiment 3 investigates the formation of circuits throughout training, and finds that circuit probing is more faithful to those circuits than linear probing. Finally, Experiment 4 analyzes a prominent language model (GPT2) on a task with an unknown causal graph (language modeling)[2].

**Circuit Probing Evaluation**   Across these four experiments, we evaluate the success of circuit probing in two ways: (1) We train a 1-nearest neighbor classifier on the output vectors produced by the discovered circuit, and then test this classifier on held-out data. If circuit probing has succeeded at finding a circuit that computes a particular intermediate variable, then its output vectors are partitioned by the possible labels of this variable, and we expect this classifier to achieve high performance. We employ a rather conservative strategy here, randomly sampling only 1 output vector for each label to train the nearest neighbors classifier. (2) We ablate the discovered circuit (i.e. invert the learned binary mask) and analyze how the model's behavior changes. This allows us to investigate whether the circuit is causal with respect to the underlying model.

---

[2]See Appendix A for all data and hyperparameter details.

### 3.1 Experiment 1: Deciphering Neural Network Algorithms

**Goal** One of the central goals of interpretability research is to characterize the algorithms that models implement Olah (2022). This lofty goal is made substantially more tractable when we can adjudicate between two hypothesized alternatives. We demonstrate that circuit probing, linear and nonlinear probing, and causal abstraction analysis all produce converging results when characterizing the algorithm implemented by a model trained on a simple arithmetic task.

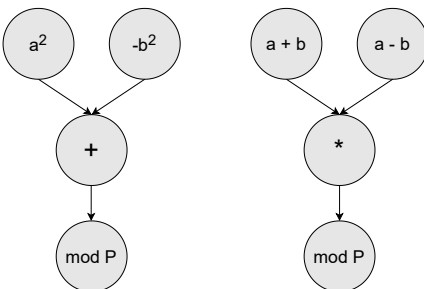

| Method | $a^2$ | $-1 * b^2$ | $a + b$ | $a - b$ |
|---|---|---|---|---|
| Circuit | 99% | 99% | 1% | 1% |
| Linear | 100% | 100% | 0% | 0% |
| Nonlinear | 100% | 100% | 1% | 1% |

Figure 2: Two hypothesized causal graphs that represent solutions to the task in Experiment 1.

Table 1: Accuracy of all probing methods for each intermediate variable in Experiment 1. All methods converge to the conclusion that the model is representing $a^2 - b^2$, rather than $(a + b) * (a - b)$.

**Task** We train a 1-layer GPT2 model to solve a task defined by the function $(a^2 - b^2)(\text{mod } P)$, where P is set to 113, and $a$ and $b$ are input variables. The input sequences are of the form $[a, b, P]$, and the model is tasked with predicting the answer based on the output embedding of the final token. All inputs are symbolic - they are one hot vectors mapping to learnable embeddings. We exhaustively generate all possible data points with $a$ and $b$ taking values $0 - 112$. Note that this input-output mapping permits at least two possible solutions, because $a^2 - b^2 = (a + b) \times (a - b)$. This is visualized by the two causal graphs depicted in Figure 2.

**Analysis Methods** We use circuit probing to determine which of the two alternative solutions the trained neural network adopts. Specifically, we search for the intermediate variables $a^2$, $(-1 * b^2)$, $(a + b)$, and $(a - b)$, all mod 113. We optimize binary masks using the output vectors from the attention and MLP blocks when they are operating on the $P$ token, which is where the final prediction is made.

We compare circuit probing to linear and nonlinear probing. We train either a linear or nonlinear classifier to map from embeddings of the $P$ token to the value of the intermediate variable that we are attempting to decode. We train counterfactual embeddings using the trained linear and nonlinear probing classifiers for this task (Tucker et al., 2021) in order to assess whether the probes are decoding information that is causally implicated in model behavior. We also compare to boundless distributed alignment search (boundless DAS; Wu et al. (2023)), a state-of-the-art causal abstraction analysis technique that is effective when one can fully specify the expected causal graph (as is the case here). Causal abstraction analysis measures whether model behavior changes as a result of interventions on intermediate variables, rather than whether the variable can be decoded directly from model representations. Thus, we can answer the same research question with causal abstraction analysis, but the results are not directly comparable to probing results (i.e., cannot be easily displayed in the same table).

**Results** We expect all methods to achieve high performance on *either* $a^2$ and $-1 * b^2$ **or** $a + b$ and $a - b$. This would adjudicate between the two alternative solutions to the arithmetic task. Indeed, we see that circuit probing, linear probing, and nonlinear probing all converge to show that the model computes $a^2 - b^2$, rather than $(a + b) \times (a - b)$ (See Table 1). With circuit probing, we can ablate the discovered circuits to ensure that they change model behavior, and thus are causally implicated in the algorithm that the model is implementing. We confirm that the circuits are causal in Appendix E. Additionally, all methods converge toward more negative results from the MLP block, indicating that the intermediate variables are computed by the attention block (See Appendix D for MLP results).

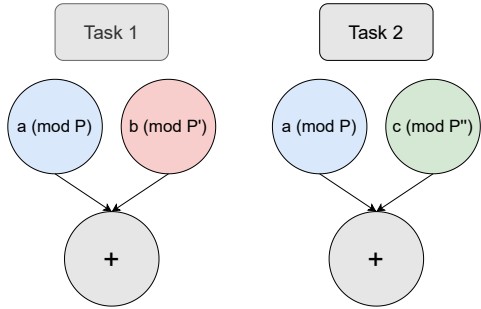

Figure 3: Causal graphs representing the Multitask dataset in Experiment 2. Note that Task 1 and Task 2 share one intermediate variable (blue) and differ by one intermediate variable (red and green).

| Train Task | Variable | Task 1 Ablation Accuracy | Task 2 Ablation Accuracy |
|---|---|---|---|
| Task 1 | Shared | 3.7% | 3.7% |
| Task 1 | Free | 3.2% | **28.1%** |
| Task 2 | Shared | 3.6% | 3.6% |
| Task 2 | Free | **48%** | 4.8% |

Table 2: Experiment 2 circuit probing ablation results. Targeted ablations of the circuits returned by circuit probing reveal a stark difference between the internal representations of the Free and Shared variables. A targeted ablation of the shared variable circuit destroys performance on both tasks, whereas a targeted ablation of Task 1's free variable harms performance on Task 1 far more (underlined) than Task 2 (bolded). The same is true in the opposite direction.

Causal abstraction analysis also supports these conclusions (See Appendix F). On the other hand, we find that counterfactual embeddings fail to elicit counterfactual behavior in all cases, and thus do not provide evidence in either direction (See Appendix G). Finally, we run a transfer learning experiment, which behaviorally demonstrates that the model is representing the task as $a^2 - b^2$, rather than $(a+b) \times (a-b)$ (See Appendix H). Overall, our results demonstrate that circuit probing agrees with existing analysis techniques when characterizing the algorithm implemented by a small model trained on a simple task.

## 3.2 EXPERIMENT 2: MODULARITY OF INTERMEDIATE VARIABLES

**Goal** We now apply circuit probing to analyze the internal organization of Transformer models, which has been the subject of several recent studies (Lepori et al., 2023b; Csordás et al., 2020; Hod et al., 2021; Mittal et al., 2022). We show that circuit probing can be used to characterize whether computations are implemented in a modular and reusable manner within a Transformer, and that nonlinear probing and counterfactual embeddings do not reveal such structure.

**Task** We train a model on a simple multitask modular arithmetic task, represented by the causal graph in Figure 3. We set $P = 29$, $P' = 31$, $P'' = 23$. The input sequences are of the form $[T, a, b, c, N]$, and the model is tasked with predicting the solution. $N$ is a separator token, and $T$ is a task token. For Task 1, we exhaustively generate all possible data points with $a$ and $b$ taking values $0 - 112$, and $c$ being a random token in the same range. Similarly for Task 2. Note that both tasks share one intermediate variable (a "shared variable"), and each task has one intermediate variable that the other does not (a "free variable").

**Analysis Methods** First, we probe for intermediate variables. We use circuit probing to probe each task individually, assessing which intermediate variables are computed when solving Task 1, and which are computed when solving Task 2. Specifically, we search for the intermediate variables $a(\mod P)$, $b(\mod P')$, and $c(\mod P'')$. We optimize binary masks using the output vectors from the attention and MLP blocks when they are operating on the $N$ token, which is where the final prediction is made. We compare circuit probing to linear and nonlinear probing. We train either a

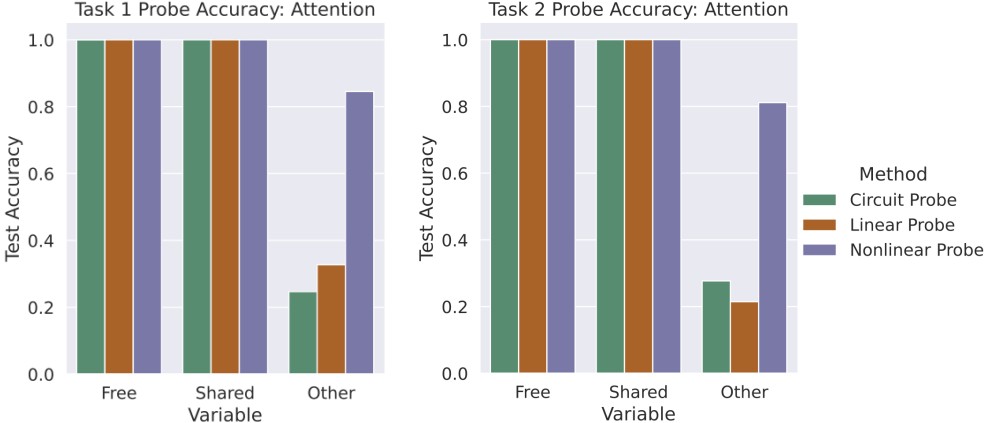

Figure 4: All probing results for Experiment 2. We find that circuit probing and linear probing generally converge to find evidence of the Free and Shared variable for both tasks, and no evidence of the irrelevant variable ("other"). On the other hand, nonlinear probing suggests that the irrelevant variable *is* represented.

linear or nonlinear classifier to map from embeddings of the $N$ token to the value of the intermediate variable that we are attempting to decode. We also run causal abstraction analysis using boundless DAS.

Next, we investigate modularity directly. We hypothesize that the model implements a *reusable* computation for the shared variable $a$(mod P) – that the same computation is used in both Task 1 and Task 2. Similarly, we hypothesize that the free variables are implemented *modularly*, – that Task 1's free variable computation can be ablated without completely destroying performance on Task 2, and vice-versa. We use targeted ablations of the circuits returned by subnetwork probing to provide evidence in support of this hypothesis. We compare against a similar analysis using counterfactual embeddings. Note that one cannot address this question using causal abstraction analysis – causal abstractions measures specific changes in model behavior, rather than investigating model representations directly.

**Results**  We first present our probing results. We expect models to compute their free and shared variables ($a$(mod P) and $b$(mod P') for Task 1, and $a$(mod P) and $c$(mod P") for Task 2), and not to compute the other variable. Our results for the attention block are shown in Figure 4. All three methods decode free and shared variables with high accuracy, indicating that the relevant variables for a given task are computed in the attention block. However, nonlinear probing (and *only* nonlinear probing) decodes the other variable with high accuracy. Causal Abstraction analysis agrees with circuit probing and linear probing (see Appendix F). This accords with prior work questioning whether expressive probes accurately reflect the causal structure of neural networks (Voita & Titov, 2020; Zhang & Bowman, 2018).

Next, we present modularity results. We expect that ablating the circuit computing the shared variable for *either* task should destroy performance across *both* tasks. On the other hand, we expect that ablating the circuit computing the free variable for Task 1 should destroy performance on Task 1 while having less of an effect on Task 2 performance (and vice-versa). In Appendix I, we analyze the morphology of the two free variable circuits returned by circuit probing and find that the circuits are only distinct in one tensor within the attention block. Thus, we only ablate the circuits within that tensor. From Table 2, we see that ablating the circuit that computes the shared variable for *either* Task 1 or Task 2 destroys performance on both tasks. On the other hand, ablating the free variable in Task 1 destroys performance on Task 1, while maintaining some performance on Task 2. Similarly for the free variable in Task 2.

We investigate the internal structure of this model using counterfactual embeddings in Appendix J, and find that counterfactual embeddings fail to provide evidence of modularity. We conclude that counterfactual embeddings act more as adversarial examples for our trained nonlinear probe, rather

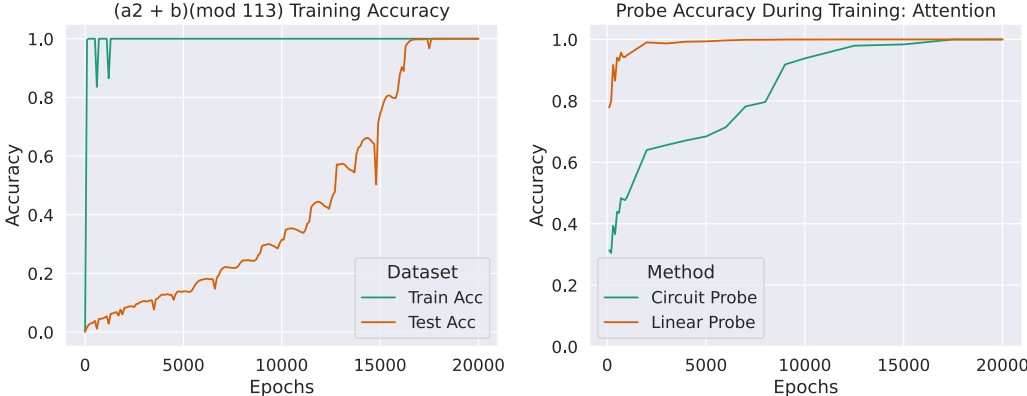

Figure 5: (Left) Experiment 3 training curve on train and test sets. We see generalization long after overfitting. (Right) Linear and circuit probing for $a^2$ throughout training. Linear probing converges to perfect accuracy very early in training, while circuit probing reveals that the circuit for $a^2$ is formed gradually through training, but achieves high performance before the overall model generalizes perfectly.

than as meaningful counterfactual inputs to the model. Overall, our results demonstrate that circuit probing is superior to competing techniques at characterizing how models structure their computations.

### 3.3 Experiment 3: Circuit Probing as a Progress Measure

**Goal**   Circuit probing allows us to gain insight into the training dynamics of Transformers at the level of intermediate variables. Recent work has shown that models may abruptly learn to generalize long after they overfit to the training data (Power et al., 2022). Despite this rather discontinuous switch from overfitting to generalization (often called *grokking*), Nanda et al. (2022) revealed that the circuit that computes the generalizable algorithm is formed continuously throughout training. While their work required reverse-engineering the entire algorithm to gain this insight into circuit formation, we reproduce their finding in a slightly different setting using circuit probing (which only requires us to hypothesize a high-level intermediate variable). In this setting, we demonstrate that circuit probing is more faithful to the circuits implemented by the underlying model than linear probing.

**Task**   We train a model on the task $(a^2 + b)(\text{mod P})$, with $P = 113$. The input sequences are of the form $[a, b, P]$, and the model is tasked with predicting the solution. Our model exhibits grokking on this task - it generalizes long after it overfits. Generalization performance increases rather slowly from epoch 0 until 10000, then rapidly climbs to near-perfect accuracy by epoch 17500. See Figure 5 (Left).

**Analysis Methods**   First, we probe for the development of the intermediate variable $a^2$ throughout training using circuit probing and linear probing. Next, we perform a *selectivity analysis* on the trained model – we probe for a variable that is not causally implicated in model behavior ($b^2$), and verify that circuit probing does not decode this intermediate variable.

**Results**   First, we investigate the development of the circuit computing $a^2$. We expect the performance of circuit probing to increase steadily throughout training, converging to a high value before the overall model generalizes. This finding would align with Nanda et al. (2022)'s finding that "circuit formation" occurs continuously, and that it completes before the overall model generalizes. We provide results from circuit probing on the attention block (See Appendix K for MLP results). From Figure 5 (Right), we see that circuit probing accuracy increases throughout training, achieving >90% accuracy before epoch 10000. Thus, we can conclude that the circuit that allows the model to generalize is formed before overall model's generalization behavior would imply. However, this circuit is not available from the outset and is developed throughout training. Linear probing tells a

markedly different story, implying that the variable required to generalize was present nearly from the beginning of training. Next, we present results from our selectivity analysis. We expect our probing methods to achieve poor accuracy when probing for $b^2$, a variable that is not causally implicated in model behavior. Circuit probing achieves 54.9% accuracy[3] at decoding $b^2$ from the fully trained model, whereas linear probing achieves 100% accuracy. See Appendix L for results throughout training. Overall these results demonstrate that circuit probing is more faithful to the underlying circuitry than linear probing.

### 3.4 EXPERIMENT 4: CIRCUIT PROBING FOR LANGUAGE MODELS

**Goal** The previous experiments focused on toy tasks in which a full causal graph could be specified. However, the reason that we are interested in developing interpretability tools is to analyze models that are used in practice, on tasks where a causal graph cannot be constructed. Here, we use circuit probing to investigate how pretrained GPT2-small and GPT2-medium perform language modeling. In particular, we investigate two linguistic phenomena that rely on syntactic number: subject-verb agreement and reflexive anaphora. Subject-verb agreement refers to the English-language phenomenon where the subject of a sentence must agree with the main verb of a sentence in syntactic number. For example: *The **keys** are on the table* is grammatical, whereas *The **keys** is on the table* is ungrammatical. We hypothesize that an intermediate variable representing the syntactic number of the subject noun is computed when predicting the main verb of a sentence. Reflexive Anaphora ensures that reflexive pronouns agree with their referents. For example: *the **consultants** injured themselves* is grammatical, and *the **consultants** injured herself* is ungrammatical. We hypothesize that an intermediate variable representing the syntactic number of the referent is computed when predicting a reflexive pronoun.

**Task** For both phenomena, we use the templates from Marvin & Linzen (2018) to generate sentence prefixes, where the continuation of the prefixes are likely to be either a main verb (when studying subject-verb agreement) or a reflexive pronoun (when studying reflexive anaphora). See Appendix M for example prefixes.

**Analysis Methods** For each phenomenon, we run circuit probing on the last word of sentence prefixes to uncover the circuit that computes the syntactic number of the subject noun or referent. We then ablate the discovered circuit, and evaluate the model's ability to grammatically continue held-out sentence prefixes. Specifically, we assess whether the model is more likely predict tokens that are consistent or inconsistent with the syntactic number of the subject/referent. For subject-verb agreement, we inspect the logits for the tokens *is* and *are* – if the logit for *is* is higher than the logit for *are* when the subject is singular (e.g. *The officer...*), then we consider the model to have succeeded on that sentence prefix. For reflexive anaphora, we run the analysis twice, once comparing the logits of *herself* and *themselves*, and again comparing the logits of *himself* and *themselves*.

Even if we recover positive results from this analysis, it could be the case that we are simply removing too many parameters from the model – rather than ablating a specialized circuit, we could be destroying the entire model. As a control, we sample 5 random subnetworks from the complement set of neurons that are in our circuit and rerun this analysis. We ensure that the random subnetworks contain the same number of parameters as our circuit in each tensor.

**Results** For both phenomena, we expect that ablating the discovered circuit will render the model worse at distinguishing the syntactic number of the subject/referent. We expect that ablating random subnetworks should not harm model performance on either dataset. We run circuit probing over all model components for both GPT2-small and medium. We present results from GPT2-small in the main body, and present GPT2-medium results in Appendix P[4]. For both datasets, we find that syntactic number is computed in Layer 6's attention block[5]. Ablating the circuit returned by circuit

---

[3]Random chance for circuit probing is 50%, as only two distinct integers in our dataset map to the same value of $b^2$(mod 113), and we are using a 1-nearest neighbor classifier.

[4]In brief, our results on GPT2-medium are largely reproduce what we find in GPT2-small, though our ablation result for subject-verb agreement is weaker.

[5]For reflexive anaphora, we note that the underlying model achieves higher accuracy when predicting the masculine pronoun. This is clear evidence of gender bias in language models, which has been well-documented elsewhere (Marvin & Linzen, 2018; May et al., 2019; Rudinger et al., 2018; Weidinger et al., 2021).

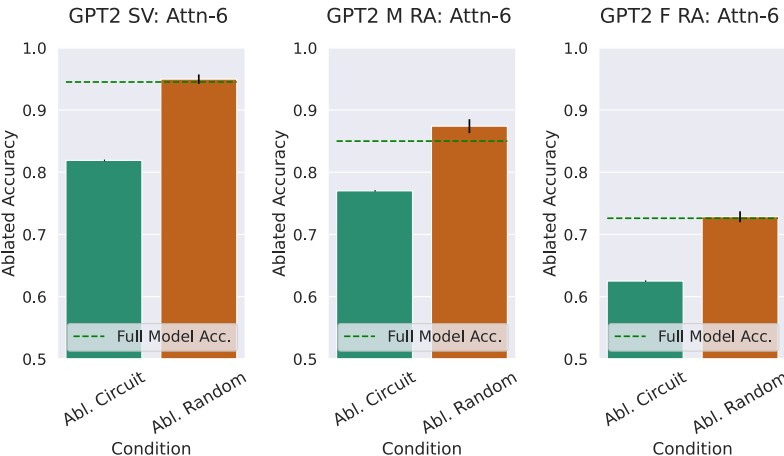

Figure 6: GPT2-Small ablation results on Layer 6's attention block. Across both Subject-Verb Agreement (Left) and Reflexive Anaphora evaluated using the masculine (Middle) and feminine (Right) pronoun, we see that ablating the discovered circuit renders the model worse at distinguishing the syntactic number of the subject/referent. Ablating randomly sampled subnetworks has little impact on the model's ability to distinguish singular and plural subjects/referents.

probing drops performance substantially for both phenomena, while ablating random subnetworks does not impact model performance (See Figure 6). Other blocks do not return this characteristic pattern as strongly (See Appendix O). This accords with prior work suggesting that syntactic dependencies are represented in middle layers of Transformers (Tenney et al., 2019; Vig & Belinkov, 2019). See Appendix N for an investigation of KNN performance, Appendix Q for an analysis of circuit overlap, and Appendix R for qualitative results.

## 4 DISCUSSION

**Related Work** Circuit probing is related to recent efforts in *mechanistic interpretability* – a burgeoning field that attempts to reverse-engineer neural network algorithms. Through substantial manual effort, researchers have uncovered the algorithms that both toy models (Olsson et al., 2022; Nanda et al., 2022; Chughtai et al., 2023) and more realistic models (Wang et al., 2022; Hanna et al., 2023; Merullo et al., 2023) are implementing. More broadly, there has been substantial work analyzing the syntactic (Linzen & Baroni, 2021; Goldberg, 2019; Tenney et al., 2018; McCoy et al., 2018) and semantic capabilities Pavlick (2022; 2023); Yu & Ettinger (2020); Hupkes et al. (2020); Dziri et al. (2023) of language models. Circuit probing is most directly related to work that attempts to decompose neural networks into functional subnetworks Csordás et al. (2020); Hamblin et al. (2022); Lepori et al. (2023b); Zhang et al. (2021); Panigrahi et al. (2023); Hod et al. (2021); Cao et al. (2021). The success of circuit probing across the diverse set of experiments presented here is further evidence that subnetworks are a useful lens through which to analyze models.

**Limitations** Circuit probing is weaker than both counterfactual embeddings and causal abstraction analysis in one key respect: it does not allow for counterfactual interventions. It is currently unknown how multiple circuits compose within a given block to create one additive update to the residual stream, so one cannot replace individual variables to elicit counterfactual behavior. Future work might seek to understand how circuits compose with one another.

**Conclusion** Through four experiments, we have demonstrated the effectiveness of circuit probing for investigating the internal representations of models. With circuit probing, one can gain insights into the underlying algorithms the model is implementing, how these algorithms are structured within the model, and how they develop throughout training. Circuit probing is more faithful to the underlying model than linear or nonlinear probing, and does not require specifying a full causal graph before testing hypotheses.

## 5 ETHICS STATEMENT

We believe that the present work is in compliance with the ICLR code of ethics. Circuit probing can be used to uncover computations that a neural network is performing. This may have future implications for bias, fairness, and safety of neural network models. However, we emphasize that the current iteration of circuit probing should not be used in isolation to assess models for social biases in real-world systems. Circuit probing can provide positive evidence that a computation is implemented, but cannot yet be used to provide evidence that a computation is definitely *not* implemented in a real-world system.

## 6 REPRODUCIBILITY STATEMENT

To foster reproducibility, we provide details on model training and hyperparameters in Appendices A and B. We provide details of our experimental design for all experiments throughout the main text, as well as in our appendices. Additionally, we provide a high-level explanation of the circuit probing algorithm in Section 2, and provide details on the loss function in Appendix C. Finally, we make our code publicly available.

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
