## A DATA AND HYPERPARAMETER DETAILS

All 1-layer GPT2 models have 4 attention heads, embedding size of 128, and MLP dimension of 512. We use the Adam optimizer (Kingma & Ba, 2014) with a learning rate of 0.001, and train with weight decay.

**Experiment 1** We train a 1-layer GPT2 model on 60% of all possible datapoints for this task, leaving the other 40% held out as a test set. For circuit probing, we train our mask on all examples from the train set (See Appendix B for those training hyperparameters) using a batch size of 500, and use updates generated from train set examples to train the 1-nearest neighbors classifier. We evaluate on the held-out test set.

For linear and nonlinear probing, we train for 100 epochs, using a learning rate of 0.1. We use this same learning rate for generating counterfactual embeddings. Our nonlinear classifier uses ReLU nonlinearity, and has a hidden size of 256.

For Boundless DAS, we train for 250 epochs, with a 2500 training examples using the Adam optimizer with a learning rate of 0.01.

For transfer experiments, we finetune on 60% of each dataset, using the Adam optimizer with a learning rate of 0.001. We train with weight decay. We train for fewer epochs with $a^2$ because the models converge very early in training.

**Experiment 2** All details are the same as in Experiment 1.

**Experiment 3** We train a 1-layer GPT2 model on 33.3% of all possible datapoints for this task, which gives us the grokking behavior that we wish to investigate. All other details are the same as Experiment 1.

**Experiment 4** We train circuit probing on 2000 examples for each dataset and test on 1000 examples, otherwise all details are the same.

## B CONTINUOUS SPARSIFICATION DETAILS

Continuous sparsification enables us to train binary masks over model weights. Our loss function is defined as:

$$\min_{m_i \in \{0,1\}} L_{soft\_neighbors}(C_{\theta \odot m_i}) + \lambda||m|| \tag{1}$$

Where $m$ is our binary mask, $C_\theta$ is a model component, with weights $\theta$, and $||m||$ is our $l_0$ regularizer, and $L_{soft\_neighbors}$ is the soft nearest neighbors loss described in Appendix C

Typically, optimizing such a binary mask is intractable, given the combinatorial nature of a discrete binary mask over a large parameter space. Instead, continuous sparsification reparameterizes the loss function by introducing another variable, $s \in \mathbb{R}^d$:

$$\min_{s_i \in \mathbb{R}^d} L_{soft\_neighbors}(C_{\theta \odot \sigma(\beta \cdot s_i)} + \lambda||\sigma(\beta \cdot s_i)||_1 \tag{2}$$

In Equation 2, $\sigma$ is the sigmoid function, applied elementwise, and $\beta$ is a temperature parameter. During training $\beta$ is increased after each epoch according to an exponential schedule to a large value $\beta_{max}$. Note that, as $\beta \to \infty$, $\sigma(\beta \cdot s_i) \to H(s_i)$, where $H(s_i)$ is the *heaviside function*.

$$H(s) = \left\{ \begin{array}{l} 0, s < 0 \\ 1, s > 0 \end{array} \right\} \tag{3}$$

Thus, during training, we interpolate between a soft mask ($\sigma$) and a discrete mask ($H$). During inference, we simply substitute $\sigma(\beta_{max} \cdot s_i)$ for $H(s_i)$. Notably, we apply continuous sparsification to a frozen model in an attempt to reveal the internal structure of this model, whereas the original

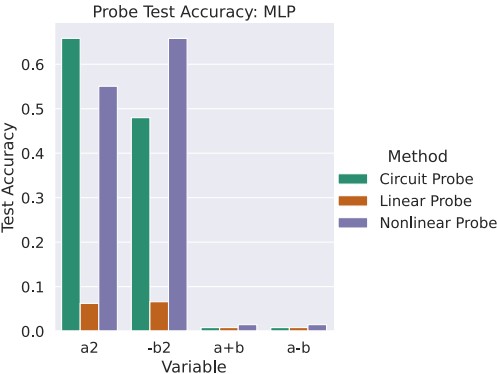

Figure 7: MLP probe accuracy for Experiment 1. All methods decode $a^2$ and $-1 * b^2$ worse in the MLP than in the attention block. Note that chance accuracy for circuit probing is effectively 50%.

work introduced continuous sparsification in the context of model pruning, and jointly trained $\theta$ and $s$.

For all experiments, we train binary masks with the Adam optimizer (Kingma & Ba, 2014), with a learning rate of 0.001. We fix $\beta_{max} = 200$, initialize the mask parameters to 0, and train for 90 epochs. The $\lambda$ parameter must scale with the number of parameters per layer. For all 1-layer GPT2 experiments, we set $\lambda = 1E - 6$. For GPT2-small and medium, we set $\lambda = 1E - 7$ and $1E - 8$, respectively.

## C    SOFT NEAREST NEIGHBORS LOSS

Given input embeddings $x$ to a model component $C$ and intermediate variable labels $y$, in a batch with $b$ samples, Equation 4 defines the full optimization objective. $\lambda$ is a hyperparameter that scales the $l_0$ regularization strength. Intuitively, this loss function pushes members of the same class towards each other, according to some distance metric, and members of different classes far from each other. Concretely, this partitions the output space of transformer layers into equivalence classes defined by the variable that we are searching for.

$$min_{m_n \in \{0,1\}} - \frac{1}{b} \sum_{\substack{i \in 1..b}} \frac{\sum_{\substack{j \in 1...b, \\ j \neq i, \\ y_i = y_j}} e^{cosine\_dist(C_{\theta \odot m}(x_i), C_{\theta \odot m}(x_j))}}{\sum_{\substack{k \in 1...b, \\ k \neq i}} e^{cosine\_dist(C_{\theta \odot m}(x_i), C_{\theta \odot m}(x_k))}} + \lambda \sum_{n \in 1...|m|} m_n \qquad (4)$$

## D    EXPERIMENT 1: MLP RESULTS

In Figure 7, we present results from Experiment 1 for all probing methods on the MLP block. First, we note that all methods perform worse at decoding $a^2$ and $-1 * b^2$. We note that chance accuracy for circuit probing is effectively 50%, whereas chance for probing methods is 0.8% (1 out of 113). Circuit probing results are generated by a 1-nearest neighbors classifier trained on the outputs of the MLP block after masking. For the variable $a^2$ and $-1 * b^2$, there are only two distinct integers that map to the same value of that variable (i.e. $4^2 (\text{mod } 113) = 111^2 (\text{mod } 113) = 4$). Because we are training the classifier with 1 vector per variable label, 50% of underlying integers are represented in the 1-NN training set. Thus, circuit probing accuracy of 50% means that the block is merely decoding the identity of the underlying token, rather than meaningfully computing an intermediate variable.

| Variable | Full Test Acc. | Ablated Test Acc. | % Parameters in Circuit |
|----------|----------------|-------------------|-------------------------|
| $a^2$    | 100%           | 0.8%              | 53.3%                   |
| $-b^2$   | 100%           | 0.9%              | 53.5%                   |
| $a+b$    | 100%           | 100%              | 0%                      |
| $a-b$    | 100%           | 100%              | 0%                      |

Table 3: Experiment 1 task performance after ablating the circuit returned by circuit probing. We see that ablating the circuit responsible for either $a^2$ or $-b^2$ destroys test accuracy. However, we see that circuit probing returns an empty circuit for both $a+b$ and $a-b$, due to $l_0$ regularization. Thus, ablating this empty circuit has no effect.

| Component | $a^2$ | $-b^2$ | $a+b$ | $a-b$ |
|-----------|-------|--------|-------|-------|
| Attn.     | 98%   | 99%    | 1%    | 1%    |
| MLP       | 2%    | 2%     | 2%    | 2%    |

| Component | Task | $a$ (mod P) | $b$ (mod P') | c (mod P") |
|-----------|------|-------------|--------------|------------|
| Attn.     | 1    | 93%         | 93%          | -          |
| MLP       | 1    | 3%          | 3%           | -          |
| Attn.     | 2    | 93%         | -            | 94%        |
| MLP       | 2    | 4%          | -            | 4%         |

Table 4: Causal abstraction analysis results for Experiment 1 (top) and Experiment 2 (bottom). Using boundless distributed alignment search, we reveal that the attention blocks in both models contain the same causal intermediate variables that circuit probing discovers.

# E  EXPERIMENT 1: CIRCUIT ABLATION RESULTS

Table 3 contains the results from running an ablation analysis on the circuits discovered in Experiment 1. We note two things: (1) ablating the circuits for $a^2$ and $-1*b^2$ destroy model performance, and (2) circuit probing returns empty subnetworks for $a+b$ and $a-b$. This is a useful feature of using $l_0$ regularization when training binary masks – if there is no signal for a given variable, circuit probing is encouraged to return a maximally sparse (i.e. empty) subnetwork.

# F  CAUSAL ABSTRACTION ANALYSIS RESULTS

In Table 4, we provide results from running causal abstraction analysis using Boundless DAS (Wu et al., 2023) on both Experiment 1 and Experiment 2. Causal abstraction analysis creates interventions on model representations in order to elicit counterfactual behavior in the downstream model. For example, for the case of $a^2 - b^2$, the model might intervene to change the value of $a^2$ to $a'^2$. The intervention is considered successful of the overall model outputs the answer to $a'^2 - b^2$. Causal abstraction analysis reports statistics in terms of the success of its counterfactual embeddings, rather than its ability to decode model representations directly. Thus, these results answer the same questions as, but are not directly comparable to, circuit probing results.

Nevertheless, these results support the results generated by circuit probing in both Experiment 1 and Experiment 2. In Experiment 1, causal abstraction analysis reveals evidence for $a^2$ and $-1*b^2$, but not $a+b$ and $a-b$. In Experiment 2, causal abstraction analysis reveals evidence that the model is using variables $a(\text{mod P})$ and $b(\text{mod P'})$ when solving Task 1, and $a(\text{mod P})$ and $c(\text{mod P"})$ when solving Task 2.

# G  EXPERIMENT 1: COUNTERFACTUAL EMBEDDINGS

We present counterfactual embedding results from Experiment 1 in Table 5. Counterfactual embeddings are embeddings that are optimized to fool a probing classifier. Formally, given a probe, $P_\theta$ trained to decode an intermediate variable, $V$, consider a residual stream state $e$ such that

| Counterfactual Embedding Success | | | | |
|---|---|---|---|---|
| Probe | $a^2$ | $-b^2$ | $a+b$ | $a-b$ |
| Linear | 100% | 100% | 100% | 100% |
| Nonlinear | 100% | 100% | 1% | 1% |
| Counterfactual Behavior Success | | | | |
| Probe | $a^2$ | $-b^2$ | $a+b$ | $a-b$ |
| Linear | 1% | 1% | 1% | 1% |
| Nonlinear | 1% | 1% | 0% | 0% |

Table 5: Experiment 1 counterfactual embedding results. (Top) Counterfactual embedding success – the percent of examples where the counterfactual optimization procedure creates an example that changes probe outputs to a particular class. We see that this optimization process largely succeeds, except for $a+b$ and $a-b$ in nonlinear probes. (Bottom) Counterfactual behavior success – the percent of counterfactual embeddings that elicit counterfactual behavior in the model. We see very poor performance on this metric, indicating that counterfactual embeddings are not producing the expected behavioral outcomes.

$P_\theta(e) = V_i$. We freeze $P_\theta$ and optimize $e$ such that $P_\theta(e') = V_j$. If $P_\theta$ is decoding information that is causally implicated in the underlying model, then replacing $e$ with $e'$ should change the output to the output one would expect from setting variable $V$ to $V_j$. We report counterfactual embedding success – the percent of embeddings that are successfully optimized to fool the probing classifier. We see that all linear probing classifiers can be fooled by counterfactual embeddings. We see that nonlinear classifiers can be fooled by counterfactual embeddings only for $a^2$ and $-1 * b^2$. Recall that all classifiers performed poorly at decoding $a+b$ and $a-b$.

Next, we analyze counterfactual behavior success – the percent of counterfactual embeddings that actually elicit counterfactual behavior in the overall model. We see that all sets of counterfactual embeddings fail to elicit counterfactual behavior. Taken in isolation, one might conclude that these probes are not decoding causally-relevant information, and thus that models are not actually computing $a^2$ and $-1 * b^2$. However, given the success of every other analysis technique at causally implicating $a^2$ and $-1 * b^2$, we may instead conclude that counterfactual embeddings are acting as adversarial examples to the probing classifier, and are destroying the embedding with respect to the underlying model.

## H   EXPERIMENT 1: TRANSFER LEARNING

To further confirm our findings in Experiment 1, we analyze whether training on $a^2 - b^2$ confers any benefits when finetuning on different tasks. In particular, we finetune the GPT2 model on a task defined by $a^2(\text{mod } 113)$, and separately on a task defined by $a+b(\text{mod } 113)$. If the model is solving the task using $a^2 - b^2$, we expect that finetuning should help the model solve $a^2$ faster than training a randomly initialized model, because the model already represents the variable necessary to solve the finetuning task. Similarly, we expect the finetuning to $a+b$ will be slower than training a randomly initialized model, because the model represents variables that are explicitly not useful for solving the finetuning task. From Figure 8, that is exactly what we see.

## I   EXPERIMENT 2: CIRCUIT OVERLAP ANALYSIS

From Figure 9, we see that the two circuits computing the free variables in Task 1 and Task 2 are largely distinct in `attn.c_attn`, but nearly completely overlapping in `attn.c_proj`. With this insight, we ablate circuit parameters just within `attn.c_attn`. We can also visualize the distribution of circuits through attention heads. In Figure 10, we see that circuit probing recovers structures that exist between particular attention heads (i.e. no single attention head is *fully* devoted to an intermediate variable) , but also partially localizes the two Free variables into specific heads (head 1 for Task 1, head 2 for Task 2).

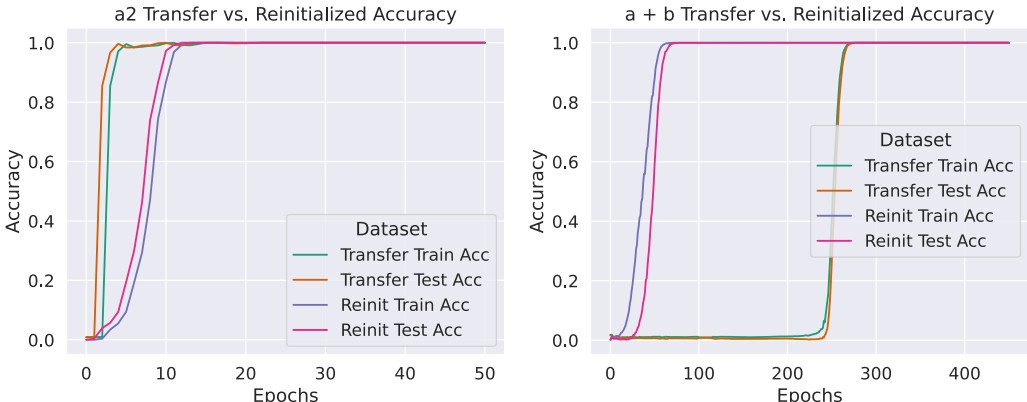

Figure 8: (Left) Transfer performance for $a^2$. We see that pretraining on $a^2 - b^2$ confers a benefit to the model when finetuning on $a^2$. (Right) Transfer performance for $a + b$. We see that pretraining on $a^2 - b^2$ is a detriment when finetuning on $a + b$.

Free Variable Circuit Overlap

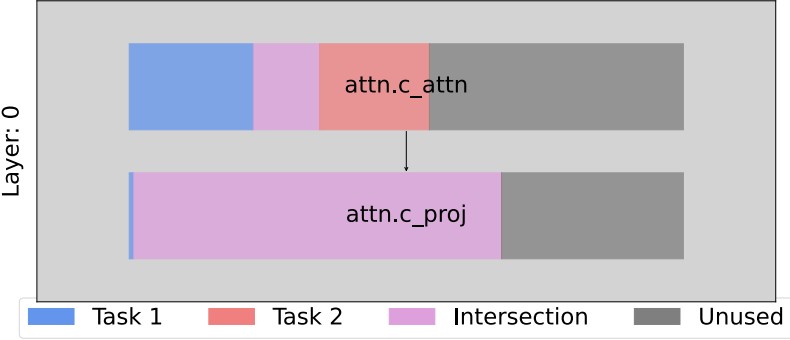

Figure 9: Experiment 2: Visualizing the distribution of circuits throughout the tensors comprising an attention block. We see that the circuits computing the free variables for Task 1 and Task 2 almost completely overlap in the c_proj tensor, but are mostly distinct in the c_attn tensor.

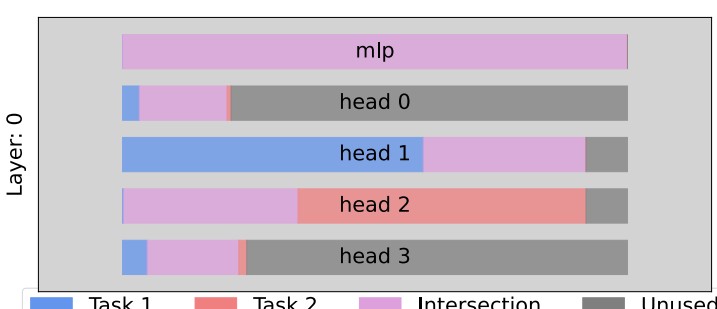

Figure 10: Circuit probing recovers some elements of known structure within Transforemrs. In particular, we see that Head 1 largely computes the Free variable in Task 1, and Head 2 largely computes the Free variable in Task 2. However, we also note that circuits extend beyond individual attention heads.

| Train Task | Test Task | Probe Var. | CE Success | Model Acc. |
|:---:|:---:|:---:|:---:|:---:|
| 1 | 1 | Free | 100% | 2.1% |
| 1 | 1 | Other | 100% | 1.8% |
| 2 | 2 | Free | 100% | 2.1% |
| 2 | 2 | Other | 100% | 2.3% |

Table 6: Experiment 2 counterfactual embedding modularity results. If the theory behind counterfactual embeddings is correct, only counterfactual embeddings optimized to change the prediction of a probe that decodes causal information should produce different output when patched into the underlying model. Concretely, counterfactual embeddings affect the "free" variable probe should result in a different prediction being made in the overall model. Counterfactual embeddings that affect the "other" variable probe should have no effect on the overall model's prediction. We see that this does not happen. Though all counterfactual embeddings succeed at changing the probe prediction (CE Sucess), they also all change the overall model prediction (CE Acc.).

## J  EXPERIMENT 2: COUNTERFACTUAL EMBEDDING MODULARITY ANALYSIS

Here, we test for modularity using counterfactual embeddings. Counterfactual embeddings are designed to reveal whether probes are reflecting information that is causally implicated in model behavior. We summarize the relevant details of their technique here, but defer to Tucker et al. (2021) for a full treatment. Given a probe, $P_\theta$ trained to decode an intermediate variable, $V$, consider a residual stream state $e$ such that $P_\theta(e) = V_i$. We freeze $P_\theta$ and optimize $e$ such that $P_\theta(e') = V_j$. If $P_\theta$ is decoding information that is causally implicated in the underlying model, then replacing $e$ with $e'$ should change the output. If this information is not causally implicated in the underlying model, then replacing $e$ with $e'$ should have no effect. For both tasks, we generate counterfactual embeddings using the nonlinear probes trained to decode the "free" and "other" variables from the residual stream after the attention block. From Table 6, we see that model performance drop to near-zero after patching in counterfactual embeddings for *either* "free" or "other" variables. This suggests that that counterfactual embeddings act more as adversarial examples to the probe, rather than providing information about causally-relevant variables.

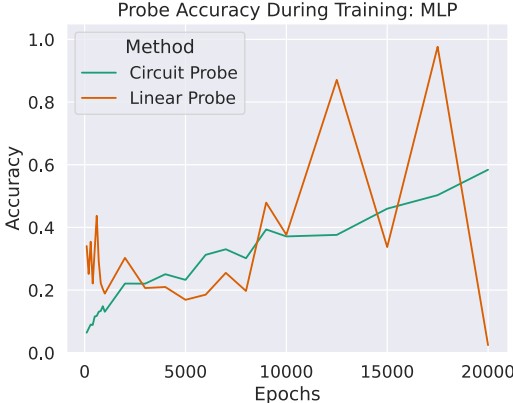

Figure 11: Experiment 3: MLP Probing results. We see chaotic results from both linear and circuit probes, indicating that the intermediate variable $a^2$ is not computed in the MLP block.

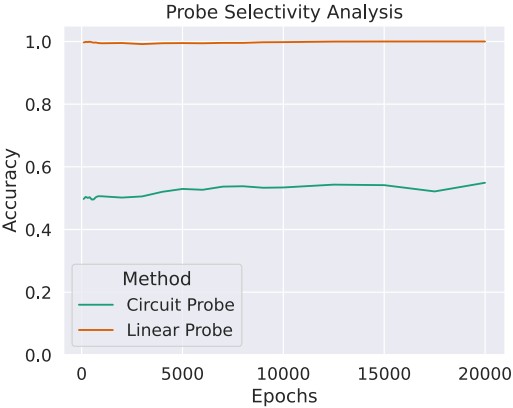

Figure 12: Experiment 3: Selectivity Analysis. Linear and circuit probing results for $b^2$ – a variable that is *not* causally implicated in the task $a^2 + b$. Circuit probing reveals that this variable is not represented at any point during training, whereas linear probing would imply that it is represented from the start of training.

## K  EXPERIMENT 3: MLP PROBE ACCURACY

Here we present the linear and circuit probe accuracy on the MLP block throughout training for Experiment 3. In Figure 11 see very messy results, further reinforcing that the intermediate variable $a^2$ is computed in the attention block.

## L  EXPERIMENT 3: SELECTIVITY THROUGHOUT TRAINING

We present results of our selectivity analysis in Experiment 3 throughout training. We see that linear probing consistently decodes the non-causal variable $b^2$, while circuit probing is consistently at chance performance. This illustrates a key problem with linear probing – it can learn to represent variables that the model does not explicitly represent.

Circuit probing results are generated by a 1-nearest neighbors classifier trained on the outputs of the MLP block after masking. For the variable $b^2$, there are only two distinct integers that map to the same value of that variable (i.e. $4^2 (\mathrm{mod}\ 113) = 111^2 (\mathrm{mod}\ 113) = 4$). Because we are training the classifier with 1 vector per variable label, 50% of underlying integers are represented in the 1-NN training set. Thus, circuit probing accuracy of 50% means that the block is merely decoding the identity of the underlying token, rather than meaningfully computing an intermediate variable.

## M    EXPERIMENT 4: LANGUAGE PREFIX EXAMPLES

Here, we present several examples of sentence prefixes for subject-verb agreement and reflexive anaphora.

Subject-Verb Agreement:

1. the farmers that the taxi driver admires (are)
2. the authors behind the assistants (are)
3. the consultant that the skaters like (is)

Reflexive Anaphora:

1. the consultants that the parents loved doubted (themselves)
2. the senators that the taxi drivers hate congratulated (themselves)
3. the mechanics thought the pilot hurt (herself).

## N    EXPERIMENT 4: GPT2-SMALL LANGUAGE KNN RESULTS

We present circuit probing KNN evaluations for both subject-verb agreement and reflexive anaphora over all model components. Recall that causal analyses indicate that a circuit that is causally implicated in computing syntactic number is located in layer 6's attention block. We note that KNN accuracy increases for every MLP block after layer 6. Because MLP blocks are applied token-wise, this suggests that the information required to decode syntactic number of both subjects and referents is present in the residual stream after this layer, but not before. See Figure 13.

## O    EXPERIMENT 4: GPT2-SMALL FULL ABLATION RESULTS

We present circuit probing ablation results for both subject-verb agreement and reflexive anaphora over all model components. In all cases, we note that ablating the circuit in attention block in layer 6 provides the greatest drop in model performance. Randomly ablating subnetworks of the same size does not harm model performance.

## P    EXPERIMENT 4: GPT2-MEDIUM RESULTS

### P.1    REFLEXIVE ANAPHORA

We present results analyzing GPT2-Medium's ability to compute the syntactic number of the referent of a reflexive pronoun. We find that the attention block in layer 7 is most causally implicated in this computation. See Figure 17. Ablating the discovered circuits harms model performance, regardless of the pronoun used for evaluation. Ablating random subnetworks of the same size does not harm model performance.

Turning to the reflexive anaphora KNN evalation, we see that the KNN accuracy of circuits trained on MLP blocks increases during and after layer 7. Because MLP blocks operate token-wise, this indicates that the information required to decode the syntactic number of referents is present in the residual stream after this layer, but not before. This strengthens our causal results analysis. See Figure 18.

For completeness, we include ablation results across all model components in Figures 19 and 20.

### P.2    SUBJECT-VERB AGREEMENT

We do not find any particular circuits that drop subject-verb agreement performance substantially when ablated (See Figure 21). This might indicate that multiple circuits across several blocks are redundantly computing the syntactic number of the subject noun.

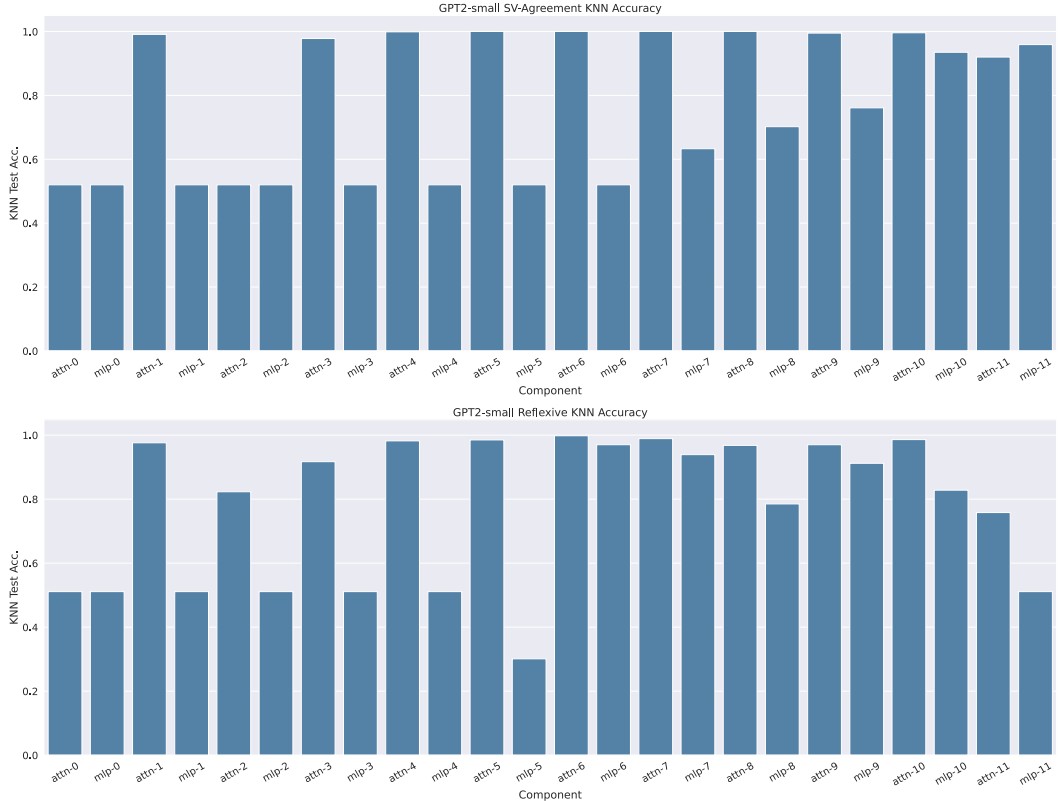

Figure 13: GPT2-Small KNN results for subject-verb agreement (Top) and reflexive anaphora (Bottom). We notice that KNN accuracy increases for MLP block after layer 6, which is where our ablation analysis located the causal circuits for both phenomena.

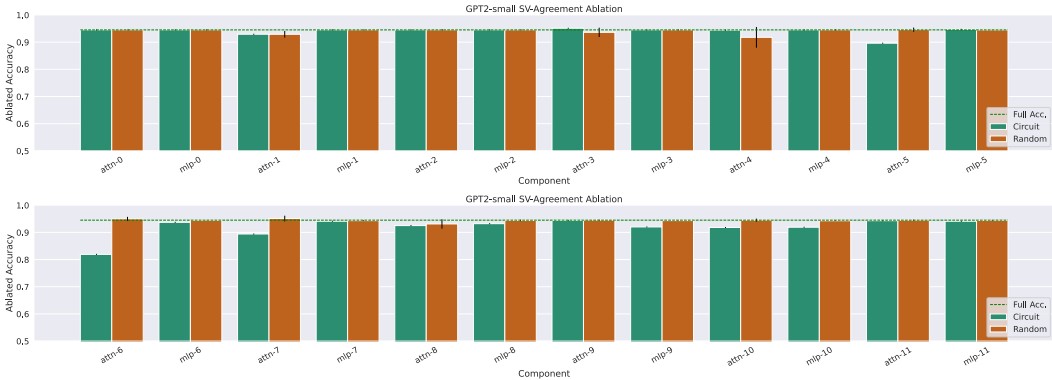

Figure 14: GPT2-Small subject-verb agreement ablation results for every model component. We note that the attention block in layer 6 provides the greatest drop in performance after ablating the discovered circuit.

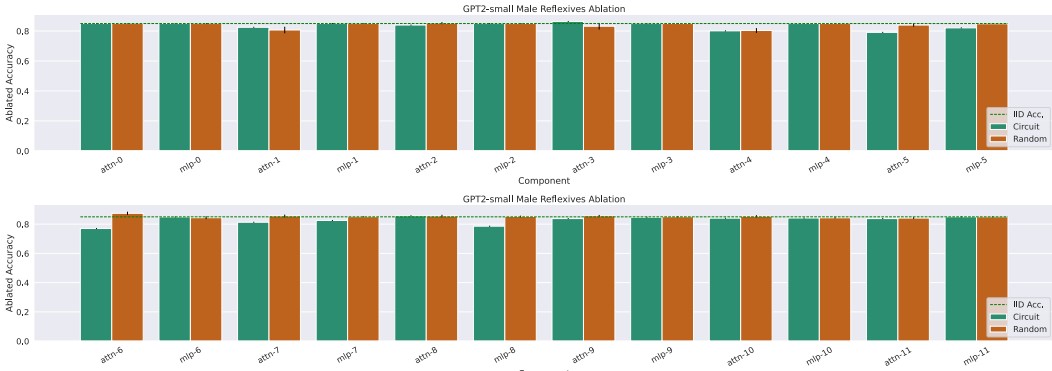

Figure 15: GPT2-Small reflexive anaphora ablation results for every model component, evaluated using the masculine pronoun. We note that the attention block in layer 6 provides the greatest drop in performance after ablating the discovered circuit.

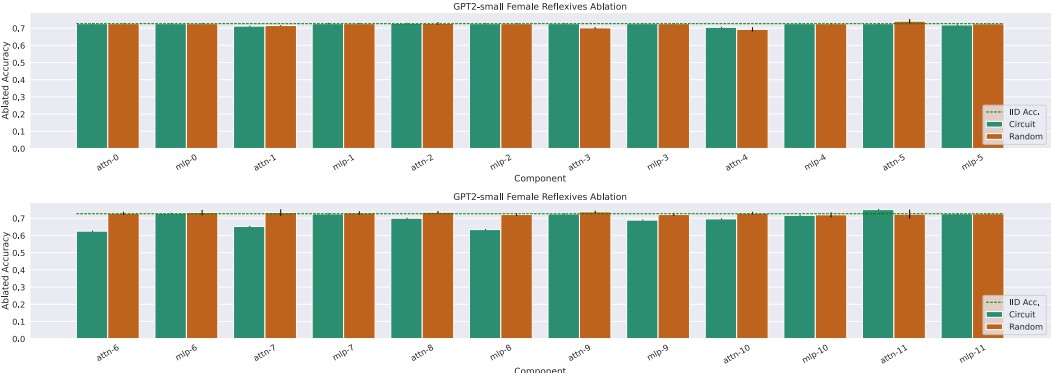

Figure 16: GPT2-Small reflexive anaphora ablation results for every model component, evaluated using the feminine pronoun. We note that the attention block in layer 6 provides the greatest drop in performance after ablating the discovered circuit.

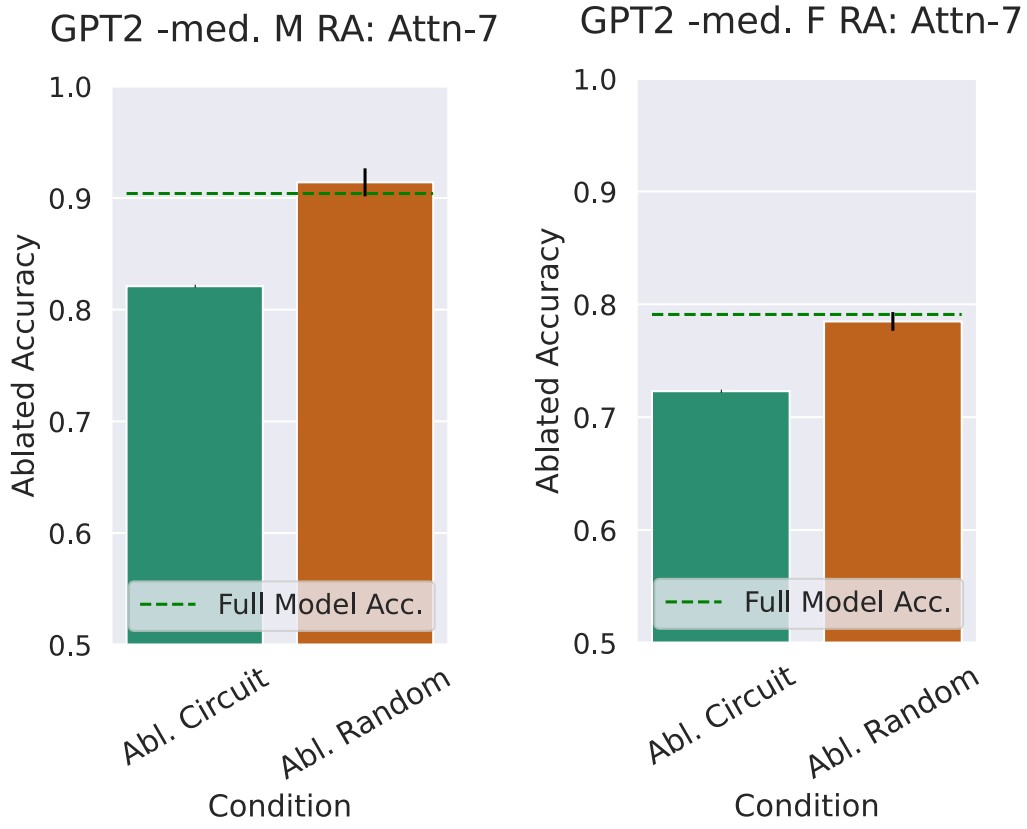

Figure 17: GPT2-Medium reflexive anaphora ablation results for the attention block in layer 7.

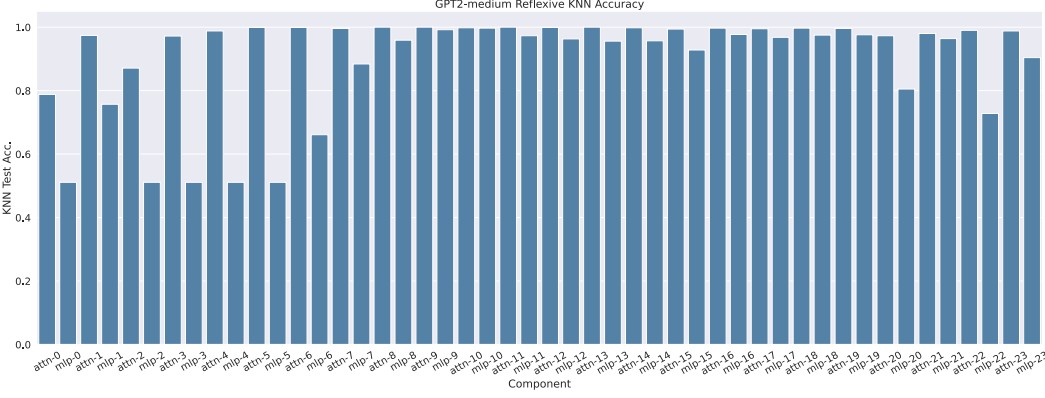

Figure 18: GPT2-Medium reflexive anaphora KNN evaluation results across all model components.

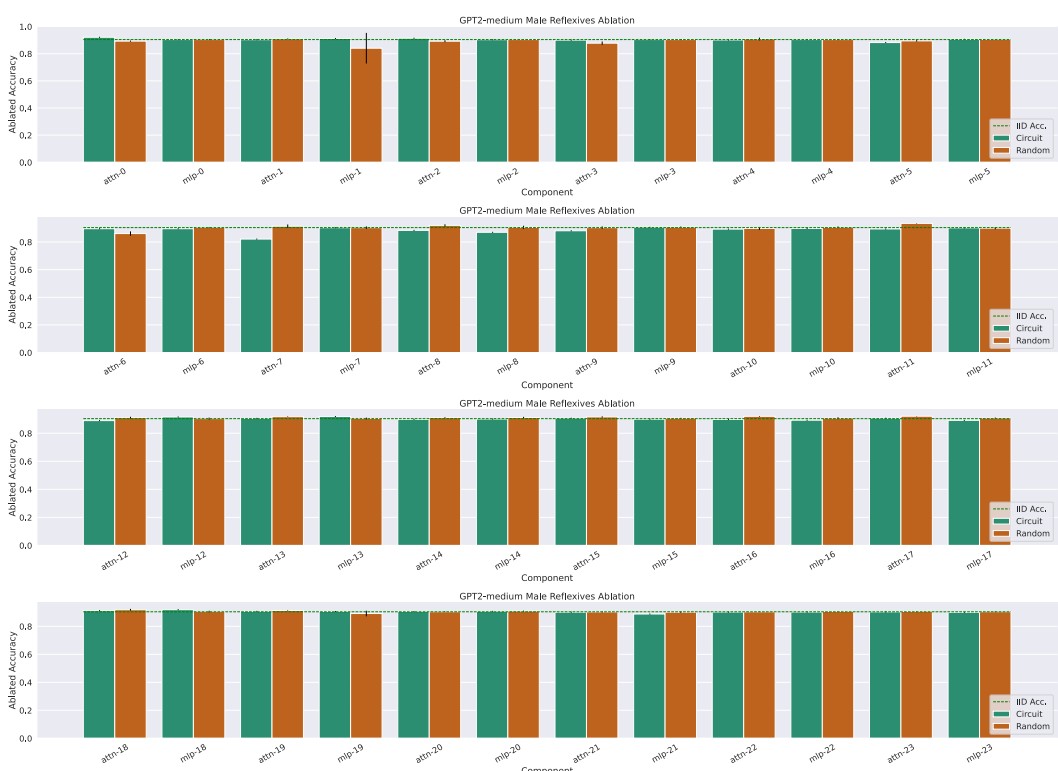

Figure 19: GPT2-Medium reflexive anaphora ablation results across all model components, evaluated using the masculine pronoun. Note that the largest drop in performance due to ablation occurs at the attention block in layer 7.

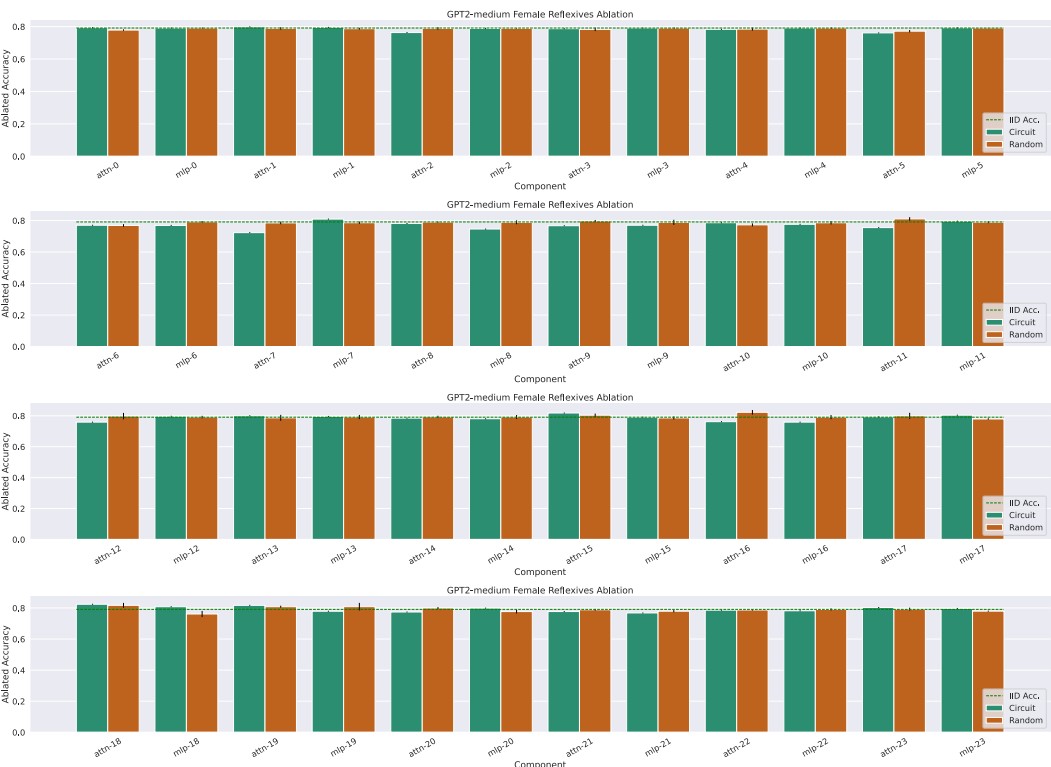

Figure 20: GPT2-Medium reflexive anaphora ablation results across all model components, evaluated using the feminine pronoun. Note that the largest drop in performance due to ablation occurs at the attention block in layer 7.

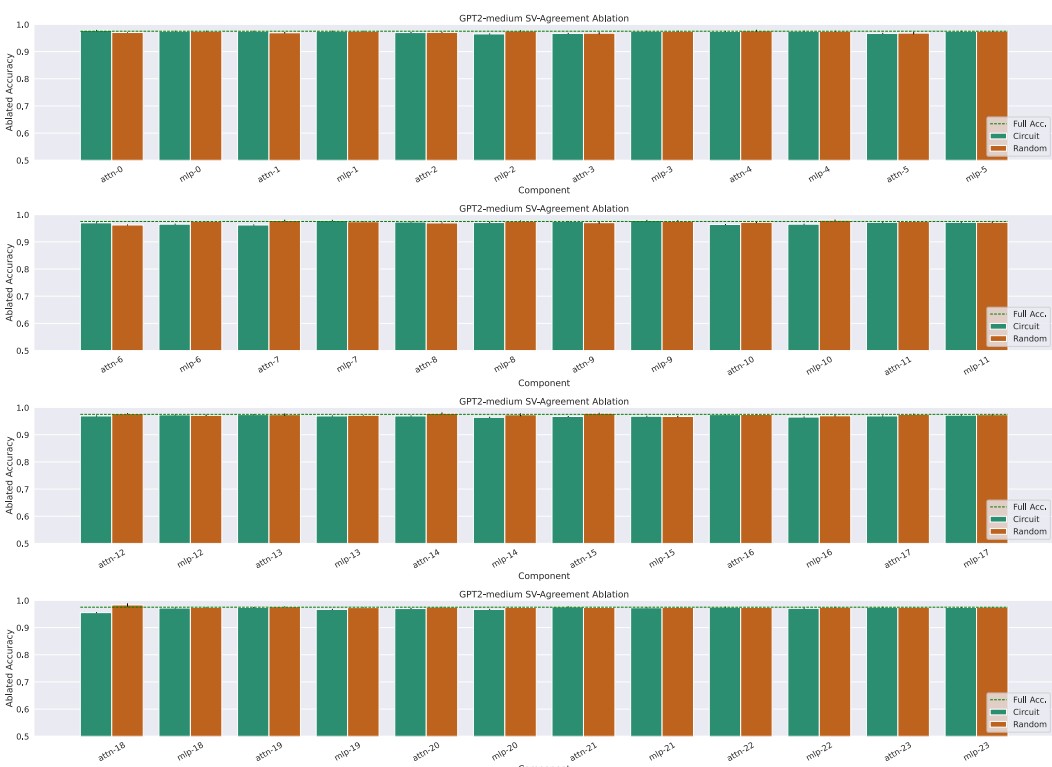

Figure 21: GPT2-Medium subject-verb agreement ablation results across all model components. These results do not implicate any specific circuit in computing the syntactic number of the subject noun.

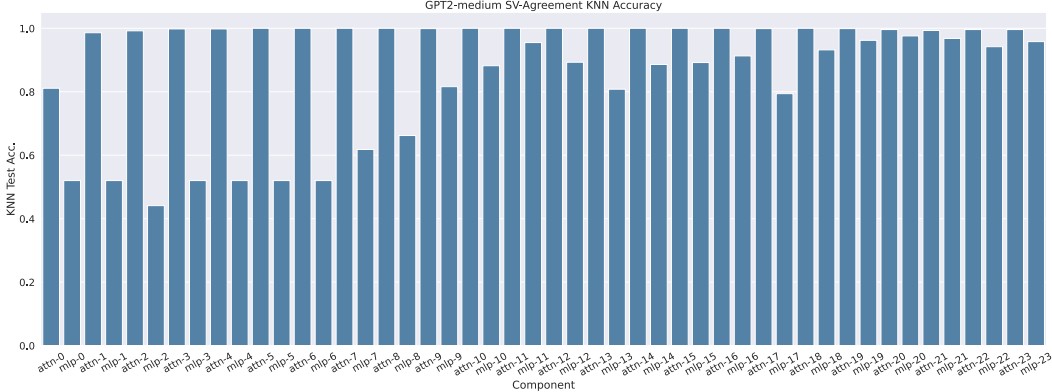

Figure 22: GPT2-Medium subject-verb agreement KNN results. We see KNN performance increase for MLP blocks around layer 7.

Turning to the subject-verb agreement KNN evalation, we see that the KNN accuracy of circuits trained on MLP blocks increases after layer 7. Because MLP blocks operate token-wise, this might indicate that the information required to decode the syntactic number of referents is present in the residual stream after this layer, but not before. However, our causal analysis does provide evidence of this. See Figure 22.

## Q    EXPERIMENT 4: CIRCUIT OVERLAP

Surprisingly, we see that there is very little overlap between the circuits used to compute the syntactic numbers of subjects and referents in GPT2-small, despite both circuits being present in the same block. See Figure 23.

## R    SUBJECT VERB AGREEMENT GPT2 MEDIUM QUALITATIVE RESULTS

We present qualitative results of ablating the subject-verb agreement circuit discovered by running circuit probing on the attention block in layer 6 of GPT2-small (See Table 7). We note that the types of tokens predicted by model qualitatively stay the same before and after ablation. This suggests that we have not destroyed the model by ablating the discovered circuit. Interestingly, we see more tokens that are explicitly consistent with the syntactic number of the subject before ablation, and fewer after ablation. This provides qualitative evidence that we have indeed ablated a circuit that was responsible for computing the syntactic number of the subject noun.

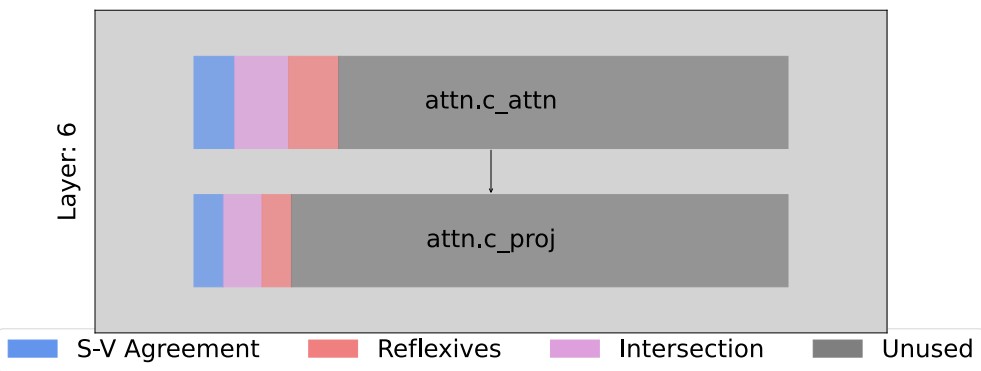

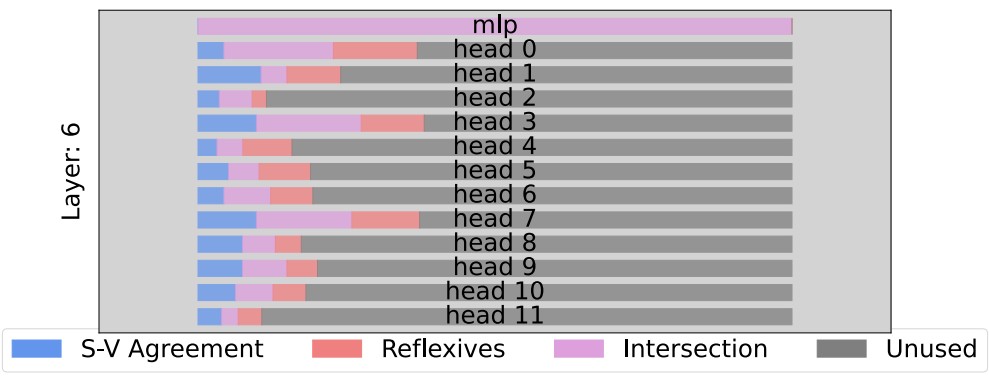

Figure 23: Circuit overlap between syntactic number and reflexive anaphora in GPT2-Small, attention block 7. We see that the discovered circuits are largely distinct. We also note that certain attention heads (0, 3, and 7) appear to be most important in computing syntactic number for both subject nouns and referents.

| Prefix | Original Output | Ablate Attn-6 Output |
|---|---|---|
| The surgeons behind the dancer | | |
| | 's | 's |
| | , | , |
| | **were** | . |
| | . | and |
| | and | _ |
| | **are** | ) |
| | said | to |
| | _ | in |
| | had | who |
| | **have** | was |
| | to | ). |
| | who | is |
| | in | said |
| | was | ), |
| | ) | **were** |
| The book from the executives | | |
| | of | of |
| | at | at |
| | , | who |
| | who | , |
| | , | and |
| | . | , |
| | and | in |
| | in | . |
| | that | to |
| | **was** | that |
| | **is** | ) |
| | to | on |
| | themselves | I |
| | on | were |
| | | are |

Table 7: Qualitative examples of the effect of ablating the circuit discovered in GPT2-small, layer 6. We record the top 15 next-token predictions. Words that are explicitly consistent with the syntactic number of the subject are bolded, words that are inconsistent are underlined.