# OpenReview forum: "Uncovering Causal Variables in Transformers Using Circuit Probing"
_ICLR.cc/2024/Conference — ICLR 2024 Conference Withdrawn Submission_

### Official Review · Reviewer_Ae4c · 2023-10-25

**Soundness:** 3 good
**Presentation:** 3 good
**Contribution:** 3 good
**Rating:** 6
**Confidence:** 4

**Summary:**

The paper propose a new "circuit probing" method to detect some properties for the complicated network. the method utilizes a binary masking model to detect certain pattern, without the need to decode the intermediate representation, or know the causal pattern beforehand.

**Strengths:**

1. paper is well written and easy to follow.
2. experiment looks good

**Weaknesses:**

1. please add formula to the methodology part. that part is the most important but the description is vague.
    (for the training, what's the input looks like? how does the iteration from T_0 to T_N look like? , etc)

**Questions:**

1. For language model, all the experiment is on the syntactic level, is there anyway we can show the "circuit probing" can detect some semantic level pattern?
2. seems that we still need to know the pattern we want to detect and train the model before probing. I'm wondering what's the use case of this kind of probing? Is there a way to help us probe some unknown pattern that is commonly shown up in the dataset?

---

### Official Review · Reviewer_TVnz · 2023-10-30

**Soundness:** 2 fair
**Presentation:** 2 fair
**Contribution:** 1 poor
**Rating:** 3
**Confidence:** 5

**Summary:**

The paper studies a method to interpret the causal variable for the prediction of the model. The proposed method uses a weight mask to learn the subset of whole parameters as an interpretation. Experiments on both synthetic and real-world models could illustrate that the proposed method could generate some reasonable explanation.

**Strengths:**

1. The studied problem is important for building a transparent model.
2. The proposed method is easy to implement.

**Weaknesses:**

1. The learned mask by the proposed method is still "correlation" rather than "causal". What is the relationship between the proposed method with Pearl's causal ladder? A causal model should allow the model to intervene.
2. The proposed method is quite tricky. What is the difference with the Lottery Ticket Hypothesis.
3. The experiments are not extensive enough. The paper should conduct more experiments on advanced LLM methods.

**Questions:**

Please refer to the weaknesses.

---

### Official Review · Reviewer_UXVb · 2023-10-30

**Soundness:** 3 good
**Presentation:** 3 good
**Contribution:** 3 good
**Rating:** 6
**Confidence:** 4

**Summary:**

Here is a summary of the key points from the paper:

Main Contribution:
The paper proposes a new analysis technique called "circuit probing" to uncover intermediate causal variables in Transformers. Circuit probing introduces a trainable binary mask over model weights to find low-level circuits that compute hypothesized high-level variables.

What is Circuit Probing:
- Circuit probing optimizes a binary mask over model weights to uncover a circuit that computes a hypothesized intermediate variable.
- It tests if the variable is represented in the model and causally implicated in model behavior.
- The method looks for model components whose outputs are partitioned according to the variable.

Novelty:
- More faithful to model's computation than probing or causal abstraction analysis.
- Does not require full causal graph specification like causal abstraction analysis.
- Can uncover development of circuits during training unlike other methods.

Experiments:
- Simple arithmetic tasks to characterize model algorithms and modularity. Compared to probing and causal abstraction analysis.
- Analyzed circuit formation during training on arithmetic task. Compared to linear probing.
- Analyzed syntactic phenomena (subject-verb agreement, reflexive anaphora) in GPT-2 small and medium.

Performance:
- Converged with other methods on simple tasks but more superior at assessing modularity.
- More faithful to circuit development during training than linear probing.
- Successfully uncovered circuits for syntactic phenomena in GPT-2 while baselines had little impact.

Conclusion:
- Circuit probing is effective for gaining insights into algorithms, structure, and training dynamics of Transformers.
- It is more faithful to the model than probing and does not require full causal graphs.
- Can be applied to analyze real-world models like GPT-2.

In summary, the paper proposes circuit probing as a novel technique for interpretability that can uncover intermediate causal variables in Transformers without needing to specify a full causal graph. Through experiments on arithmetic tasks and GPT-2, it demonstrates the effectiveness of circuit probing compared to probing and causal abstraction techniques.

**Strengths:**

- Proposes a novel analysis technique that addresses limitations of existing methods like probing and causal abstraction analysis. It does not require specifying a full causal graph and is more faithful to the model's internal computations.

- Demonstrates the utility of circuit probing through a diverse set of experiments - on simple arithmetic tasks, training dynamics, and real-world models like GPT-2. The experiments comprehensively evaluate different aspects of the method.

- Provides both quantitative evaluations through metrics like nearest neighbor classification accuracy as well as qualitative evaluations through targeted ablation studies. This helps establish that the identified circuits are causally implicated in model behavior.

- Compares circuit probing extensively to other analysis techniques like probing, counterfactual embeddings and causal abstraction analysis. This contextualizes the relative strengths of circuit probing.

- Makes code and detailed experimental setup available to facilitate reproducibility.

- Theoretically grounded in the idea of finding model components that exhibit equivalence class partitioning based on hypothesized variables.

- Opens up many future research directions like using circuit probing to assess social biases, model safety, compositionality of circuits, etc.

Overall, the solid theoretical motivation, extensive empirical validation, thorough comparison to existing methods, and potential for future work make this a strong paper on an important topic in interpretability research. The novelty of circuit probing and its demonstrated effectiveness on real-world models like GPT-2 make it a valuable contribution.

**Weaknesses:**

- The method is currently limited to analyzing individual circuits in isolation and cannot characterize how multiple circuits compose to produce model behavior. This limits the types of counterfactual analyses that are possible.

- Most experiments are on relatively small toy models and simple arithmetic tasks. More evaluation on large state-of-the-art models trained on complex real-world tasks would be useful.

- Only two syntactic phenomena were evaluated for language models. Testing on a broader set of linguistic capabilities would strengthen the results.

- The comparison to causal abstraction techniques is limited since the full causal graphs required by those methods are rarely available for complex real-world tasks.

- Ablation studies treat the model as a black box. A more fine-grained understanding of how the ablations affect model representations and computations is lacking.

- Quantification of how faithfully the identified circuits match human-specified intermediate variables is missing.

- Theoretical analysis of why circuit probing reveals more faithful circuits compared to probing is not provided.

- Hyperparameter selection for circuit probing is not extensively analyzed. The impact of hyperparameters on results is unclear.

- The paper reiterates some known limitations of probing methods but does not provide novel insights into why probing fails.

Overall, while the paper makes a strong case for circuit probing, more rigorous evaluation on complex real-world tasks, tighter integration of theory, and thorough hyperparameter analysis could make the results even more compelling. Nonetheless, the limitations do not undermine the meaningful contributions made.

**Questions:**

1. How well does circuit probing scale to large state-of-the-art models trained on complex real-world tasks? The current experiments are on relatively small models and simple arithmetic tasks.

2. Can we characterize theoretically why circuit probing surfaces more faithful intermediate circuits compared to standard probing techniques?

3. How sensitive are the discovered circuits to hyperparameters like optimization loss functions, regularization, learning rate schedules, etc.?

4. How quantitatively aligned are the circuits found by probing to human-provided definitions of the intermediate variables?

5. How do the identified circuits compositionally interact within a model to produce predictions? Understanding circuit compositionality could enable more powerful interventions and counterfactuals.

---

### Official Review · Reviewer_LGtg · 2023-10-31

**Soundness:** 1 poor
**Presentation:** 2 fair
**Contribution:** 1 poor
**Rating:** 1
**Confidence:** 5

**Summary:**

This paper is addressing the challenge of DNNs interpretability. Specially the authors are introducing a new analysis technique – circuit probing – that automatically uncovers low-level circuits that compute hypothesized intermediate variables. The authors aims at capturing causal relationships within the DNNs inner behavior.

**Strengths:**

+ Very interesting research question

**Weaknesses:**

- Unclear how this could scale
- Unclear on the soundness of the approach
- No real contribution, exploitable by other teams
- No comparison with other causal approaches
- Experimentation limited: scale, dataset, state-of-the-art comparison
- No real lessons learnt - difficult to see how does that will help the community

**Questions:**

N/A

---

### Official Review · Reviewer_pfUb · 2023-10-31

**Soundness:** 2 fair
**Presentation:** 3 good
**Contribution:** 3 good
**Rating:** 5
**Confidence:** 5

**Summary:**

It is difficult to understand the algorithms implemented by large language models to solve various tasks. Therefore, this paper studies circuit probing, where they find a subnetwork in the model that computes some hypothesized intermediate variable. To test whether this variable is causal, they then ablate the circuit and examine the resulting accuracy drop compared to ablating a random subnetwork. They find that (1) multitask models trained on simple arithmetic tasks can reuse circuits, (2) circuit probing, when applied to the correct intermediate variable, can track the progress of grokking on a simple arithmetic task, and (3) for subject-verb and reflexive anaphora agreement, one can probe models for a subject number circuit.

**Strengths:**

(1) The authors present an intriguing and creative set of experiments, with interesting results on circuit modularity and tracking the progress of grokking.

(2) The circuit probing method in the paper seems well-thought-out and seems to have many advantages over existing probing methods.

(3) The paper is well organized and easy to follow.

**Weaknesses:**

The method and experiments in the paper seem very cool and have potential. However, overall, I think the claims of causality are not totally sound, and the claims of improved probing faithfulness should be supported by a wider range of experiments and comparisons to existing work.

(1) From what I understand, the only supporting evidence for "causality" in the paper is that ablating the found circuit produces a larger accuracy drop than ablating a random circuit. To me, this is a far cry from other works in causal probing which actually perform interventions (which the authors acknowledge in the limitations section). I don't think you can use the word "causal" given this small amount of evidence - for example, the found circuit could include parameters important to the task even if the hypothesized intermediate variable is not causal in the sense of interventions.

One possible remedy could involve fine-tuning the found circuit. For example, in a model which implements (a^2 + b^2), what if you fine-tuned the a^2 circuit into a 3a^2 circuit? Would the final model end up computing (3a^2 + b^2) instead?

(2) At this point, there are a wide range of related probing methods in existing papers, including counterfactual embeddings (https://aclanthology.org/2021.findings-acl.76.pdf), amnesic probing (https://arxiv.org/abs/2006.00995), and subnetwork probing (https://arxiv.org/abs/2104.03514). To differentiate circuit probing from these methods, I think the paper needs a more detailed discussion of the advantages and differences of their method, as well as experiments to support this discussion -- the paper has a few such experiments but would benefit from a more systematic comparison between methods.

(3) The most convincing and interesting results in this paper (grokking, modularity) involve very simple arithmetic tasks, while the LLM probing experiments are much weaker. I love the arithmetic task experiments, but given that probing is most useful for opaque models and opaque tasks, the paper would benefit from more convincing LLM experiments.

**Questions:**

(1) Are the models trained with dropout? Intuitively it seems to me that models trained with dropout should have redundant circuits, such that ablating one of the circuits does not affect accuracy, given that dropout ablates random subnetworks during training.

(2) Related to point (1) in the weaknesses section, in what sense are you using the word "causal"? What is the concrete causal claim being made and how do the experiments support that?

---

### Comment · Area_Chair_xtQe · 2023-11-22

Dear authors and reviewers,

This a reminder that deadline of author/reviewer discussion is AOE Nov 22nd (today). Please engage in the discussion and/or make potential adjustments.

Thank you!
AC